# Genome-Wide Analysis of the RAV Gene Family in Wheat and Functional Identification of TaRAV1 in Salt Stress

**DOI:** 10.3390/ijms23168834

**Published:** 2022-08-09

**Authors:** Yun-Xin Luo, Shou-Kun Chen, Peng-Dan Wang, De Peng, Xu Zhang, Hai-Feng Li, Cui-Zhu Feng

**Affiliations:** 1State Key Laboratory of Crop Stress Adaptation and Improvement, School of Life Sciences, Henan University, Kaifeng 475000, China; 2College of Agronomy, Northwest A&F University, Yangling, Xianyang 712100, China

**Keywords:** wheat, transcription factor, RAV, salt stress, gene expression

## Abstract

RAV transcription factors (TFs) are unique to higher plants and contain both B3 and APETALA2 (AP2) DNA binding domains. Although sets of RAV genes have been identified from several species, little is known about this family in wheat. In this study, 26 RAV genes were identified in the wheat genome. These wheat RAV TFs were phylogenetically clustered into three classes based on their amino acid sequences. A *TaRAV* gene located on chromosome 1D was cloned and named *TaRAV1*. *TaRAV1* was expressed in roots, stems, leaves, and inflorescences, and its expression was up-regulated by heat while down-regulated by salt, ABA, and GA. Subcellular localization analysis revealed that the TaRAV1 protein was localized in the nucleus. The TaRAV1 protein showed DNA binding activity in the EMSA assay and transcriptional activation activity in yeast cells. Overexpressing *TaRAV1* enhanced the salt tolerance of Arabidopsis and upregulated the expression of *SOS* genes and other stress response genes. Collectively, our data suggest that TaRAV1 functions as a transcription factor and is involved in the salt stress response by regulating gene expression in the SOS pathway.

## 1. Introduction

Wheat (*Triticum aestivum*) is one of the main staple crops grown worldwide [1,2]. The growth and development of wheat are adversely influenced by abiotic stresses, e.g., salt, drought, and cold [3]. Transcription factors (TFs), such as APETALA2/ETHYLENE RESPONSIVE FACTOR (AP2/EREBP, AP2/ERF), basic region/leucine zipper (bZIP), MYB (contained varying numbers of MYB domain repeats conferring their ability to bind DNA), and WRKY (contained conserved DNA-binding domain which named WRKY domain) TF families, play important roles in the plant stress responses, regulating downstream genes through binding to the DNA elements in the promoters of target genes [4,5].

The RAV (Related to ABI3/VP1) transcription factor family is plant-specific, and most of its members contain two DNA-binding domains, the B3 and AP2 domains [6]. The RAV proteins could be assigned to the B3 or the AP2/EREBP superfamilies [6]. The B3 transcription factors contain the B3 DNA binding domain [7], which was recognized as the third basic domain in the maize transcription factor VIVIPAROUS1 (VP1) [8]. *B3* genes have functions in green algae, mosses, liverworts, ferns, and gymnosperms [9]. The B3 superfamily comprises several subfamilies, including LAV (Leafy Cotyledon2 [LEC2]-Abscisic Acid Insensitive3 [ABI3]-VAL), ARF (Auxin Response Factor), RAV, and REM (Reproductive Meristem) families [10]. The AP2/EREBP family transcription factors can regulate plant development, hormonal transduction, and stress responses and are classified into four subfamilies: DREB (Dehydration Responsive Element-Binding), ERF (Ethylene Responsive Element Binding Protein), AP2 (APETALA2) and RAV (Related to ABI3/VP) [11,12,13]. The two different DNA-binding domains bestow RAV proteins more functions.

Six RAV TFs have been identified in *Arabidopsis*, all containing both the AP2 and the B3 DNA binding domains [6]. In rice, fifteen genes have been identified as *RAV*, among which only four, *OsRAV8*, *OsRAV9*, *OsRAV11,* and *OsRAV12*, encode proteins containing both AP2 and B3 domains [14]. Like in rice, thirteen genes have been identified as *RAV* in soybean, and only five encode proteins containing both domains [15]. In recent years, *RAV* genes have also been identified in other plant species: 33 in cotton, 11 in pear, and six in loquat. [16,17,18].

AtRAV1 and AtRAV2 were the first two RAV proteins reported [19]. The AP2 and B3 domains of AtRAV1 could bind to CAACA and CACCTG motifs, respectively [19]. AtRAV1 plays important roles during plant development, hormone signaling, and stress responses. Overexpressing *AtRAV1* suppresses lateral roots and rosette leaves and promotes leaf sentence [20,21]. AtRAV1 negatively regulates ABA signal transduction by directly repressing the transcription of *ABI3*, *ABI4*, and *ABI5.* [22]. AtRAV1, AtRAV1L, and AtRAV2 (TEM2) act as positive regulators during drought and salt stress responses [23,24]. AtRAV2 (TEM2) regulates the expression of *FLOWERING LOCUS T* (*FT*) and the GA4 biosynthetic gene *AtTEM1* [25,26]. *RAV* genes in other species have also been identified as involved in stress responses or plant development processes. For example, GmRAV negatively regulates photosynthesis and plant growth, with its overexpression lines showing less growth in leaves, roots, and stems [27]. Overexpression of another *RAV* gene, *GmRAV1*, enhances soybean’s salt and drought tolerance [15]. MtRAV3 plays an important role in salt and osmotic stress response and plant growth [28]. The salt and osmotic tolerance could be enhanced by overexpressing *BpRAV1* in birch plants [29], and *CaRAV1* in *Arabidopsis* [30,31]. However, overexpression of *GhRAV1* reduced the resistance to salt and drought in *Arabidopsis* [32].

RAV TFs also play important roles during the biotic stress response. Tomato RAV2 (SlRAV2) could activate *PR* gene’s expression and function in disease resistance [33]. CaRAV1 could enhance the biotic pathogen resistance in *Arabidopsis* [30]. MeRAV1 and MeRAV2 play important roles in disease resistance against cassava bacterial blight and oxidative stress and drought response in cassava [34,35,36]. During plant development, FaRAV1 activates the expression of anthocyanin pathway genes and positively regulates anthocyanin accumulation in strawberries [37]. ZmRAVL1 could alter the endogenous brassinosteroid content and leaf angle by regulating the expression of *Bri1* [38]. The sugarcane RAV transcription factor plays an essential role during aerenchyma formation by regulating endopolygalacturonase expression [39]. Although they have been studied in various species, the number and functions of wheat RAV TFs are still unclear.

According to the above reports, we made the hypothesis that RAV transcription factors play important roles for stress responses in wheat. We thus studied the number, sequence, and structure of wheat *RAV* genes. Furthermore, we cloned one that is located on chromosome 1D and named it *TaRAV1*. We found that the expression of *TaRAV1* was slightly induced by heat but strongly inhibited by salt, ABA, and GA. The TaRAV1 protein was localized in the nucleus and showed both DNA binding and transcriptional activation activities. Overexpressing *TaRAV1* in *Arabidopsis* increases the transcription of the *SOS* genes and *RD29A*, *RD29B,* and *RD22*, enhancing salt tolerance.

## 2. Results

### 2.1. Identification and Phylogenetic Analyses of Wheat RAV Genes

From searching the wheat genome, we identified a total of 26 *RAV* genes. These genes were unevenly distributed on nine chromosomes (Figure 1) and named *TaRAV1A* to *TaRAV11D* according to homology and their locations on chromosomes. Among them, 10 *TaRAV* segmental duplication genes were identified and formed seven segmental duplication pairs; three tandem duplication pairs were generated from six *TaRAV* genes (Figure 1a and Appendix A). Each TaRAV protein contained a conserved AP2 domain, a nuclear localization signal, and a B3 domain (Appendix A). The protein length ranged from 305 (TaRAV10D) to 429 (TaRAV1A) amino acids, and the molecular weight from 33.099 kDa (TaRAV10D) to 47.63 kDa (TaRAV1A) (Appendix A). Protein subcellular localization was predicted with the CELLO web server, showing that 17 TaRAVs were localized in the nucleus, eight in chloroplasts, and one (TaRAV9B) in mitochondria ().

Using program MEGA 7, we analyzed the sequence alignment of 26 wheat, four rice, and six *Arabidopsis* RAVs, and constructed an un-rooted neighbor-joining (NJ) phylogenetic tree [6,14]. According to the clade support values, these RAVs could be classified into five groups (Figure 1b). Group I was the largest, containing 13 wheat RAVs, while groups AtRAV-I and AtRAV-II contained *Arabidopsis* RAVs only.

The genomic DNA sequence and CDS (coding sequence) were analyzed to reveal the gene structure. The *TaRAV* genes are similar: all contained only one exon, except for *TaRAV8D*, which contained two exons (Appendix A).

### 2.2. Expression Patterns of TaRAV Genes in Different Tissues

To explore the function of *TaRAV* genes, we used qRT-PCR to analyze their expression in different tissues. As shown in Figure 2, 10 genes are highly expressed in roots, and six (*TaRAV1B*, *TaRAV1D*, *TaRAV3D*, *TaRAV6A*/*U*, *TaRAV7A*/*B/D*, and *TaRAV11D*) showed a high expression level in stems. Some homoeologous wheat genes showed similar expression patterns. For example, *TaRAV9A*, *TaRAV9B*, and *TaRAV9D* are mainly expressed in roots. *TaRAV1A*, *TaRAV1B*, and *TaRAV1D* are weakly expressed in leaf and inflorescence tissues.

### 2.3. Expression Patterns of TaRAV1

According to the phylogenetic analysis (Figure 1), TaRAV3D was close to OsRAV11 (LOC_Os01g49830) and OsRAV12 (LOC_Os05g47650). Its coding sequence showed 70% and 72% homology with *AtRAV1* (At1g13260) and *AtRAV1L* (At3g25730), respectively. *AtRAV1* has been functionally characterized in our previous study [22]. All five proteins contain a conserved AP2 domain and a B3 domain (Appendix A). Therefore, we chose *TaRAV3D* for further analysis and renamed it *TaRAV1*.

The qRT-PCR method was used to determine the expression levels of *TaRAV1* in different organs. As shown in Figure 2, *TaRAV1* was moderately expressed in roots, leaves and inflorescences, and highly expressed in stems (Figure 2). Stress treatment assays showed that *TaRAV1* was up-regulated by heat but down-regulated by salt, ABA, and GA treatment. No significant difference was found in the *TaRAV1* expression levels after cold and PEG treatments (Figure 3a). We also analyzed the expression level of *TaRAV1* at different time points under NaCl treatment. When treated with 200 mM NaCl for 0.5 h, the expression of *TaRAV1* decreased significantly. The expression level was lowest after 1-h treatment and recovered partly with longer treatments. These data indicate that *TaRAV1* responds to abiotic stress, e.g., heat and salt, and hormone transduction, e.g., ABA and GA.

### 2.4. TaRAV1 Is Localized in the Nucleus

As a putative transcription factor, the TaRAV1 protein contains a predicted nuclear localization signal (NLS) between the two DNA binding domains (Appendix A). To investigate its subcellular localization, we constructed the *pSUPER:TaRAV1-GFP* vector and transformed it into tobacco (*Nicotiana tabacum*) leaves. Hence, the nucleus located protein marker, AtH2B-mCherry [40] was involved to detect the nucleus localization of TaRAV1. Observation with confocal microscopy found that TaRAV1-GFP fluorescence was well co-localized with AtH2B-mCherry and clearly in the nucleus of epidermal cells, suggesting that TaRAV1-GFP was localized in the nucleus (Figure 4 upper level and Appendix A). The *S**UPER* promoter driving *GFP* construction (*pSUPER:GFP*) was introduced as a negative control. When co-expressed with AtH2B-mCherry in tobacco leaves, the GFP was determined at cytoplasm of the epidermal cells, yet the AtH2B-mCherry was determined only in the nucleus (Figure 4, lower level). These data demonstrate that TaRAV1 is localized in the nucleus.

### 2.5. TaRAV1 Shows Both Transcription Activation and DNA Binding Activities

To test the transcription activation activity, we expressed TaRAV1 in yeast strain *AH109*, in which *HIS3* was used as a reporter. As shown in Figure 5a, the yeast cells carrying the *pGBKT7-TaRAV1* plasmid grew well on a histidine-deficient medium, similar to the positive control carrying *pGA**BKT7-RecT*, while the vector control (*pG**BKT7*) could not grow. These results indicate that TaRAV1 transcriptionally activated the reporter gene *HIS3* in yeast cells.

To determine the DNA binding capability of TaRAV1, we expressed and purified the TaRAV1-GST fusion protein from *E. coli* and carried out an electrophoretic mobility shift assay (EMSA). Since the TaRAV1 protein was predicted to contain AP2 and B3 DNA binding domains, which bind to cis-elements CAACA and CACCTG, respectively, a biotin-labeled probe containing both elements was used for the test [19]. The results showed that TaRAV1 could bind the DNA probe containing both DNA elements but failed to bind the probe containing mutant DNA elements (Figure 5b) [8], indicating that TaRAV1 can specifically bind to the two cis-elements.

### 2.6. Overexpressing TaRAV1 in Arabidopsis Enhances Salt Tolerance

To investigate the function of TaRAV1 in plants, the *TaRAV1* CDS driven by the *Super* promoter was transformed into *Arabidopsis*. Three transgenic lines (L2, L3, and L6) were chosen for the testing (Figure 6a). We found that when grown on regular MS media, the *TaRAV1* transgenic lines showed no difference from wild-type control (WT), while they showed obvious growth advantages on media containing NaCl. The root length of the transgenic seedlings under NaCl treatment was positively correlated with the expression levels of *TaRAV1* (Figure 6b). When grown on media containing 120 mM NaCl, the roots of *TaRAV1* transgenic lines L2, L3, and L6 were longer than those of WT by 26.3%, 49.4%, and 69.1%, respectively. When grown on 150 mM NaCl, the effect of transgene on root length became more dramatic; the roots of *TaRAV1* transgenic lines L2, L3, and L6 were longer than those of WT by 41.4%, 58.4%, and 117.7%, respectively (Figure 6c). In addition, we tested the sensitivity of *TaRAV1* transgenic lines to stress hormone ABA but found no differences between the transgenic lines and WT (Appendix A). These results demonstrate that TaRAV1 could enhance salt tolerance in *Arabidopsis*.

### 2.7. The Expression of Salt-Responsive Genes Is Altered in TaRAV1 Transgenic Arabidopsis

Using the qRT-PCR method, we examined the expression levels of several salt-responsive genes, including the genes involved in the SOS signaling and the stress response genes *RD29A*, *RD29B*, and *RD22* [41,42,43,44,45]. The results showed that the expression levels of all these genes were induced by NaCl treatment. Compared with those in WT, the expression levels of *SOS1*, *SOS2,* and *SOS3* in *TaRAV1* transgenic lines were slightly upregulated under normal conditions, but increased significantly after salt stress treatment, especially in L3 and L6 lines. The expression levels of *AtRD29A*, *AtRD29B,* and *AtRD22* were higher in *TaRAV1* transgenic lines than in WT after salt stress treatment but showed no difference under normal conditions (Figure 7). These results suggest that TaRAV1 could enhance the expression of those genes related to salt responses.

## 3. Discussion

### 3.1. Wheat Has More RAV Genes Than Diploid Plant Species

In the wheat genome, we identified twenty-six *RAV* genes, more than those identified in *Arabidopsis* (6), rice (4), and soybean (6) [6,14,15], and this may reflect that wheat is an allohexaploid with a huge genome [46]. This large number of RAV genes may be one of the essential reasons why wheat is more tolerant to abiotic stresses such as drought and extreme temperatures. A more complex genome can contain more stress response and resistance genes and increase stress tolerance. We propose that RAV is one of the gene families involved in stress response and resistance, which may be the reason it has more family members in wheat. In addition, gene duplication contributed to the expansion of the wheat *RAV* gene family, further explaining why wheat has more *RAV* genes than other space.

### 3.2. TaRAV1 May Work as a Transcription Factor in Wheat

Our research revealed that TaRAV1 protein showed 70% sequence homology with AtRAV1. Like other RAV proteins, TaRAV1 contains one AP2 domain, one B3 domain, and nuclear localization signals (NLSs) in between (Appendix A). TaRAV1-GFP fusion protein is localized in the nucleus of leaf epidermal cells when expressed in tobacco leaves (Figure 4). TaRAV1 has transcriptional activity in yeast cells (Figure 5a). EMSA shows that TaRAV1 binds to the canonical cis-elements CAACA and CACCTG [19] (Figure 5b). These results suggest that TaRAV1 has the characteristics of transcription factors and may act as a transcription factor in wheat.

### 3.3. TaRAV1 Plays an Important Role in Salt Response

Previous publications have reported that the expression of several *RAV* genes was related to stress responses [6]. For example, the expression of *AtRAV1* is down-regulated by ABA, brassinosteroid, and touch-related stimuli in a dose-dependent and biphasic manner [20,22,47]. The expression of *CaRAV1* is upregulated by ABA, NaCl, and cold [30]. The expression of *GhRAV1* [32], *BnaRAV-1-HY15* [48], *MtRAV3* [28], and *BpRAV1* [29] is upregulated by ABA, NaCl, or osmotic agents PEG or mannitol. In addition, the expression of *GmRAVs* in soybean changes in response to drought, NaCl, and ABA [27]. Our results showed that the expression of *TaRAV1* slightly increased under heat stress and was down-regulated by ABA, NaCl, and GA treatment, with no obvious change observed under PEG and cold treatments (Figure 3). These observations indicate that RAV transcription factors function in stress responses in different plant species.

The *TaRAV1* transgenic Arabidopsis plants showed an improved salt tolerance (Figure 6), consistent with previous observations that overexpressing *AtRAV1*, *AtRAV2,* and *AtRAV1L* enhanced the tolerance to salt and osmotic stresses in *Arabidopsis* [23] and in cotton [24]. Overexpression of *GhRAV1* reduced salt and drought tolerance in *Arabidopsis* [32], while overexpression of *CaRAV1*, *BpRAV1,* and *MtRAV3* could enhance osmotic stress tolerance in *Arabidopsis*, birch, and *Medicago truncatula*, respectively [28,29,30]. According to these data, we propose that RAV transcription factors play important roles in plant response to stresses, especially salt stress, and overexpressing *RAV* genes could improve plants’ capability to acquire resistance.

Salinity involved genes, such as *SOS* genes, and abiotic stress involved genes, such as *RD29A*, *RD29B,* and *RD22*, were examined when we studied the function of TaRAV1 (Figure 7). Previous studies indicated that the expression of *SOS1*, *RD29A*, *RD29B*, and *RD22* was induced by salt stress [43,44,45]. Our analysis showed that the expression of the three *SOS* genes was enhanced in *TaRAV1* transgenic lines either under normal conditions or under NaCl treatment (Figure 7), indicating that TaRAV1 affects the SOS pathway in *Arabidopsis*. Furthermore, the transcriptional expression levels of *RD29A*, *RD29B*, and *RD22* were increased in *TaRAV1* transgenic lines under NaCl treatment (Figure 7), suggesting that TaRAV1 participates in the salt-stress response by regulating the expression of stress involved genes. These results demonstrated that TaRAV1 acts as a positive regulator of the salt stress response.

## 4. Materials and Methods

### 4.1. Identification and Phylogenetic Analysis of RAVs

Genomic DNA sequences, coding sequences (CDSs), cDNA sequences, and protein sequences of *Triticum aestivum* (IWGSC), *Oryza sativa*, and *Arabidopsis thaliana* were obtained from the Ensembl Plants database [49]. To identify the wheat RAVs, three steps were used. First, two hidden Markov models (HMMs) of the AP2 (PF00847) and B3 (PF02362), downloaded from the Pfam database [50] and set as the query to search against wheat genome protein sequence using HMMER 3.0 with an e-value < 10^−5^. Second, the RAVs of rice and Arabidopsis were blasted against the putative RAVs in the first step with an e-value < 10^−5^ [6,14]. Third, NCBI-CDD [51] and SMART [52] were used to validate the putative RAVs, and sequences without AP2 and B3 domains were removed. After that, the wheat RAVs were obtained.

The ExPASy web server [53] was used to predict the molecular weight of RAVs. The CELLO web [54] server was used to predict the subcellular localization. The chromosome location information of wheat was obtained from the Ensembl Plants database [49]. Gene duplication information was obtained by the MCScanX program with default parameters [55]. Chromosome distribution and gene duplication were visualized using the Circos v0.69 program [56]. An unrooted neighbor-joining (NJ) tree was constructed and visualized by MEGA 7 software with 1000 bootstrap replications based on the full-length protein sequence alignment of wheat, rice, and Arabidopsis RAVs [57].

### 4.2. Plant Materials and Growth Conditions

The seeds of *T. aestivum* (Chinese Spring) were surface-sterilized with 75% (v/v) ethanol for 1 min and 1% NaClO for 15 min, followed by washing with sterile water. Then, the sterilized seeds were planted in an artificial climate chamber at 26/22 °C (day/night) with a photoperiod of 14/10 h (day/night). For the analysis of different tissues, the leaves, roots, stems, and inflorescences were collected at the heading stage. For stress response analysis, two-week-old seedlings of *T. aestivum* (Chinese Spring) were treated with heat (42 °C), cold (4 °C), 20% PEG6000, 200 mM NaCl, 100 μM ABA, or 100 μM GA solution for 2 h. For salt stress at different time points, two-week-old seedlings of *T. aestivum* (Chinese Spring) were treated with 200 mM NaCl for 0, 0.5, 1, 2, 3, 5, 7, and 12 h. After treatment, whole plants were selected for RNA extraction and further qRT-PCR testing. For stress response analysis of transgenic Arabidopsis, seven-day-old wild-type and transgenic Arabidopsis seedlings grown on MS medium were transferred to MS medium with 150 mM NaCl for 3 and 6 h and then selected for RNA extraction.

The *TaRAV1* transgenic lines were generated by cloning the coding sequence of *TaRAV1* into the *pCAMBIA1300* vector and transferred into wild-type Arabidopsis plants (Columbia ecotype) [58]. Wide-type and transgenic Arabidopsis seeds were sterilized with 1% NaClO solution for 10 min, followed by washing with sterile water. The sterilized seeds were germinated on MS medium with or without NaCl under 14 h light/8 h dark at 22 °C for 8 d. Then, the phenotype was compared, and the root length was measured.

### 4.3. Quantitative Real-Time PCR Analysis

For quantitative real-time PCR analysis, cDNA was synthesized with an HiScrip III RT SuperMix Kit for QPCR (R323-01, Vazyme). Quantitative real-time PCR was performed using ChamQ Universal SYBR qPCR Master Mix (Q711-02, Vazyme) on a Q7 Real Time PCR System (Thermo Fisher Scientific) following the manufacturer’s protocol. Relative quantitative results were calculated by normalization to *Atactin* or *Taactin*. The fluorescence signal was obtained during the PCR annealing step. The primers used are listed in Appendix A.

### 4.4. Electrophoretic Mobility Shift Assay (EMSA)

EMSA was conducted using a Light Shift Chemiluminescent EMSA Kit (Pierce) according to the manufacturer’s protocol. The recombinant TaRAV1-GST fusion protein was purified from *E. coli*. The DNA sequence containing both AP2 and B3 DNA binding elements was labeled with biotin and used as the EMSA probe (Figure 5b). The same sequence without biotin was used as a competitor, and sequences with mutations in DNA elements were used as a negative control.

### 4.5. Subcellular Localization Assay

In order to determine the subcellular localization of TaRAV1, the *pSUPER:TaRAV1-GFP* construct was generated and co-transferred with *35S:AtH2B-mCherry* into tobacco (*Nicotiana tabacum*) leaves for subcellular localization assays with confocal laser scanning microscopy (Zeiss LSM 980 with Airysca, Germany). AtH2B is located in nucleus [40]. For GFP signal detection, the argon laser excitation source was 488 nm, and the detection filters ranged from 500 to 585 nm. For mCherry signal detection, the argon laser excitation source was 543 nm, and the detection filters ranged from 570 nm.

### 4.6. Transcriptional Activity Assay

For the transcriptional activity assay, the *TaRAV1* CDS was fused to GAL4 Bind Domain in the *pG**BKT7* vector. The *pG**BKT7-TaRAV1* constructs were transformed into yeast cells. The transformed yeast cells were diluted with a 10× dilution series and dotted on SD/-Trp plates. The transcriptional activity of TaRAV1 was judged by the growth of yeast cells on SD/-Trp/-Ade/-His plates. The *pG**BKT7* vector was transformed into yeast cells as a negative control. *pG**BKT7-RecT* was transformed into yeast cells as a positive control.

## 5. Conclusions

Here, 26 RAV TFs were identified in wheat, which contained both AP2 and B3 domain. *TaRAV1* is located on chromosome 1D, and its expression level was slightly induced by heat but strongly inhibited by salt, ABA, and GA. The TaRAV1 protein showed nucleus location, DNA binding activity in vitro, and transcriptional activity in yeast. The overexpression of *TaRAV1* could enhance the salt tolerance of Arabidopsis.

## Figures and Tables

**Figure 1 ijms-23-08834-f001:**
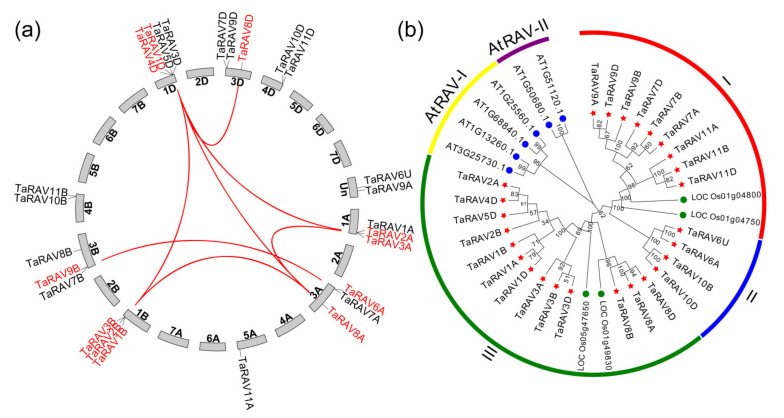
Chromosome location and Phylogenetic analysis of *RAV* genes in wheat. (**a**) Chromosome location, and segmentally duplicated gene pairs of wheat (*Triticum aestivum*) *RAV* genes; (**b**) Phylogenetic tree of 26 wheat RAV TFs, 6 *Arabidopsis* RAV TFs, and 4 rice RAV TFs.

**Figure 2 ijms-23-08834-f002:**
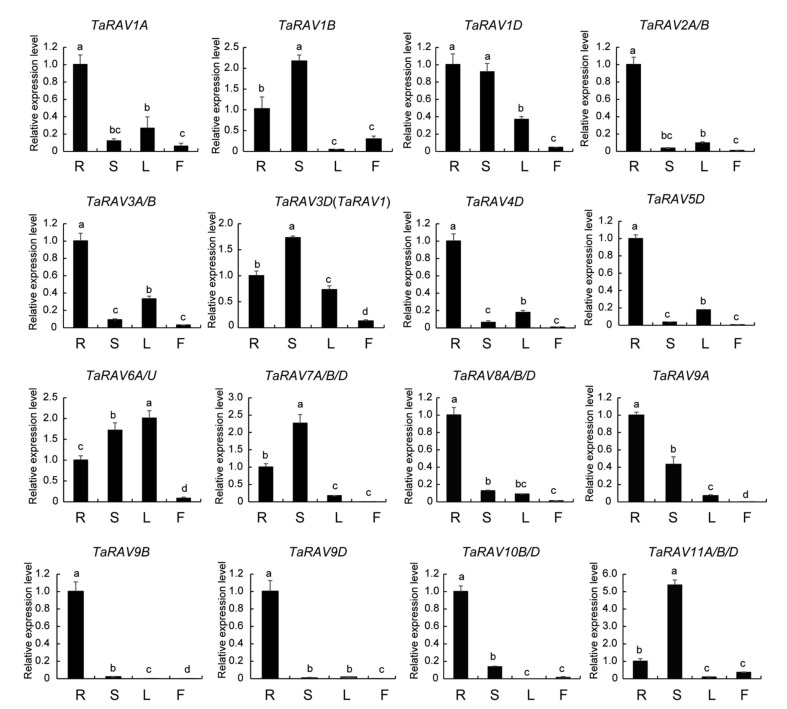
Expression levels of *TaRAV* genes in different wheat tissues.RNA was extracted from the root (R), stem (S), leaf (L), and inflorescence (F) tissues, and analyzed using the qRT-PCR method. Transcript levels of *TaRAV* genes were normalized to that of *Taactin*, and the levels of root samples were set to 1. Data shown are means ± SE, n = 3. Different letters represent significant differences (*p* < 0.05, one-way ANOVA).

**Figure 3 ijms-23-08834-f003:**
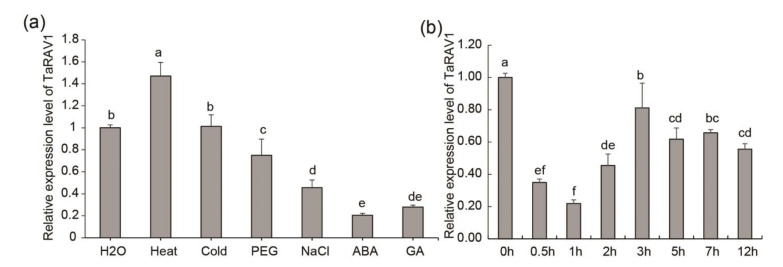
qRT-PCR analysis of *TaRAV1* expression in wheat seedlings treated with stresses or hormones. (**a**), Expression levels of *TaRAV1* in response to different treatments. Seeds were germinated under normal conditions, and before RNA extraction, 2-week-old seedlings were treated for 2 h with heat (42 °C), cold (4 °C), PEG (20% PEG6000), NaCl (200 mM), ABA (100 μM), or GA (100 μM). (**b**), Expression levels of *TaRAV1* in response to different lengths of NaCl treatment. Seedlings were prepared the same way as in (**a**), and treated with 200 mM NaCl for different lengths of time. Transcript levels were quantified using the qRT-PCR method, and normalized to that of *Taactin*. Data shown are means ± SE (n = 3). Different letters represent significant differences (*p* < 0.05, one-way ANOVA).

**Figure 4 ijms-23-08834-f004:**
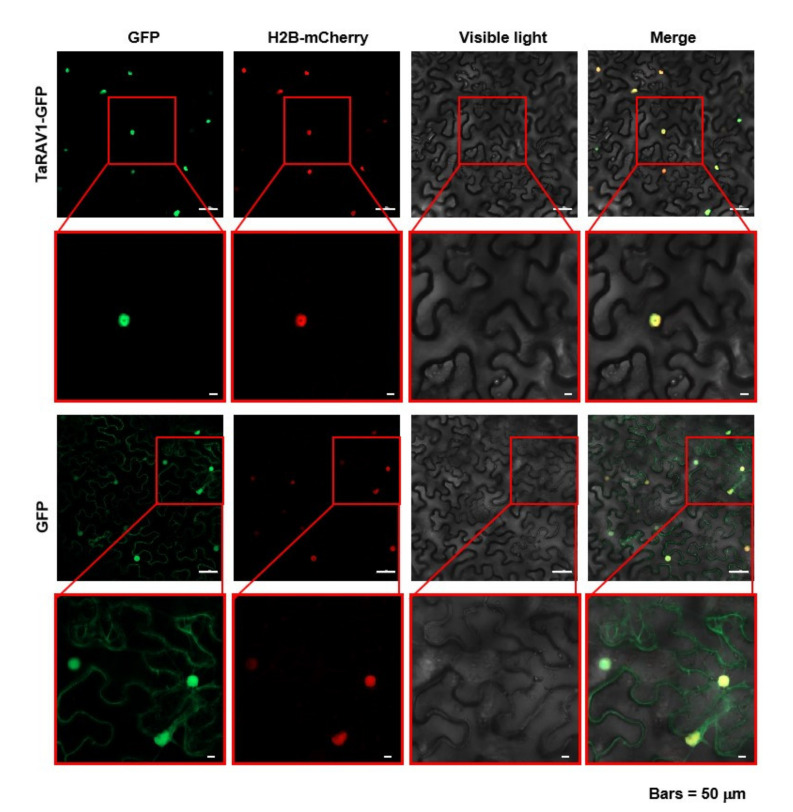
Subcellular localization of TaRAV1 protein. AtH2B-mCherry is a marker for nucleus localization. Constructs *pSUPER:TaRAV1-GFP* and *35S:AtH2B-mCherry* were co-transformed into tobacco leaves using the Agrobacterium infiltration method. *pSUPER:GFP* and *35S:AtH2B-mCherry* were co-transformed into tobacco leaves as a negative control. GFP and mCherry signals were detected by confocal microscopy 3 days after infiltration. Scale bars, 50 μm.

**Figure 5 ijms-23-08834-f005:**
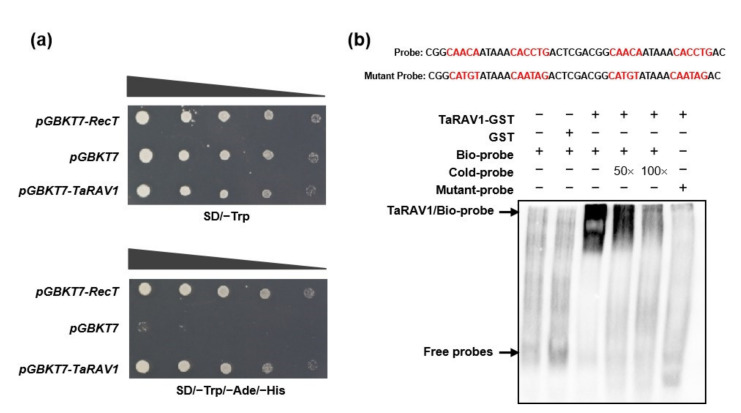
Transcription activation and DNA binding activities of TaRAV1. (**a**), Growth of yeast cells expressing TaRAV1 on selection media. Plasmid *pGBKT7-TaRAV1* was transferred into Yeast *AH109*, and transformants were spotted on SD/−Trp medium (SD minimal medium lacking Trp, upper panel) and SD/−Trp/−Ade/−His medium (SD minimal medium lacking Trp, Ade and His, lower panel). Yeast cells harboring *pGADT7-RecT* were used as a positive control; yeast cells harboring *pGBKT7* empty vector were used as a negative control. (**b**), EMSA DNA binding assay. GST-tagged TaRAV1 protein was expressed and purified from *E. coli*. The probes containing the canonical cis-elements (CAACA and CACCTG) or mutated elements were labeled with biotin for incubation with TaRAV1-GST protein. Unlabeled probes were added for the competition assay. GST protein was used as a negative control.

**Figure 6 ijms-23-08834-f006:**
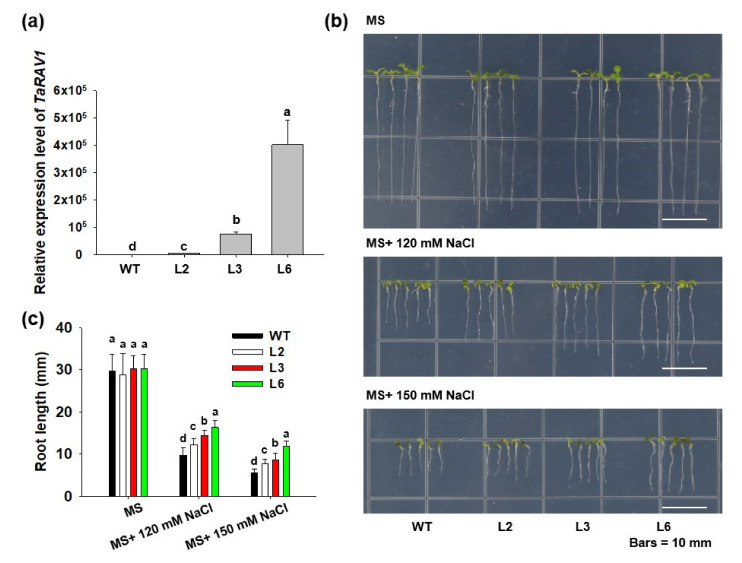
Effect of overexpressing *TaRAV1* on Arabidopsis salt tolerance. (**a**), *TaRAV1* expression level in wild-type (WT) and 3 selected transgenic lines (L2, L3, L6). Seedlings grown on MS media for 14 days were collected for RNA extraction. Transcript levels of *TaRAV1* were quantified using the qRT-PCR method and normalized to that of *Atactin*. The level of WT is set to 1. Data shown are means ± SE (n = 3). (**b**), Root elongation of WT and transgenic seedlings on MS agar media. WT and *TaRAV1* transgenic seeds were germinated on MS media, and 6-day-old seedlings were transferred to MS media or MS media containing 120 or 150 mM NaCl (MS + 120 or 150 mM NaCl). Plates were cultured vertically for 8 days before measuring root length. Scale bars, 10 mm. (**c**), Measurement of root length of the seedlings shown in b. Data shown are means ± SE (n ≥ 40). Different letters represent significant differences (*p* < 0.05, one-way ANOVA).

**Figure 7 ijms-23-08834-f007:**
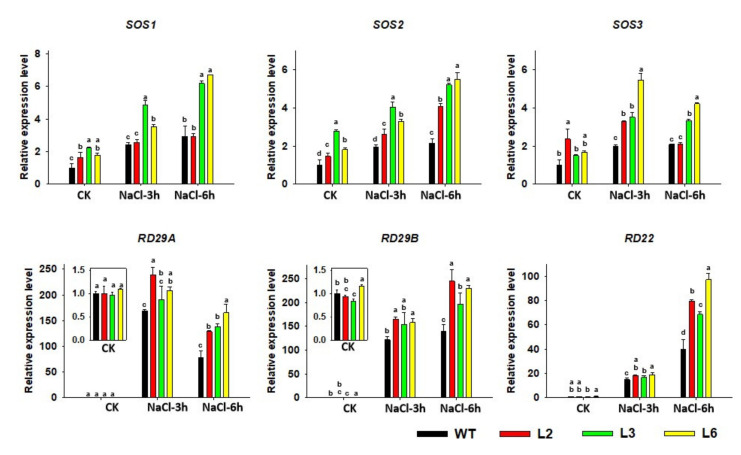
Expression of salt-responsive genes in *TaRAV1* transgenic Arabidopsis. The 7-day-old seedlings of WT and 3 transgenic lines (L2, L3, L6) were treated with 150 mM NaCl for 3 and 6 h. Transcript levels of selected genes were quantified using the qRT-PCR method, and normalized to *Atactin* mRNA. The transcript levels of WT under control condition (CK) were set to 1. Data shown are means ± SE (n = 3). Different letters represent significant differences (*p* < 0.05, one-way ANOVA).

## Data Availability

Not available.

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
