# Peer review of "Genome-Wide Analysis of the RAV Gene Family in Wheat and Functional Identification of TaRAV1 in Salt Stress"

_ijms, 2022, doi:10.3390/ijms23168834_

Round 1

Reviewer 1 Report

Work on "Genome-Wide Analysis of the RAV Gene Family in Wheat and Functional Identification of TaRAV1 in Salt stress" by Luo et al. is extremely interesting, but requires a few minor changes:

- please remove the yellow markings from the text, because I think these are leftovers after editing the text and they are not needed here;

- please check the text for an explanation of the abbreviations used, whether they are first developed everywhere, e.g. abbreviations used in lines 43-44 "The B3 43 superfamily comprises several subfamilies, including LAV, ARF, RAV, and REM [10]" .

- figure 2, please explain the abbreviations used in the graphs in the description (it is about R, L, S, F);

In summary, the work is well written and extremely interesting.

Reviewer 2 Report

Dear Authors,

I again have opportunity to assess the article “Genome-Wide Analysis of the RAV Gene Family in Wheat and Functional Identification of TaRAV1 in Salt stress” indicated the presence of new factors in slat stress. However article has one little problem: The statistical analyses of Figure 2, 3 and 7 is problematic. Authors must decide to use one way of markings of statistical significance not two because it is confusing. I suggest to use different small letters to mark significance. Authors updated all other matters according mu suggestions and raised quality of paper.

Sincerely,

Author Response

This manuscript is a resubmission of an earlier submission. The following is a list of the peer review reports and author responses from that submission.

Round 1

Reviewer 1 Report

Dear Authors,

I had great opportunity to review manuscript entitled: “Genome-Wide Analysis of the RAV Gene Family in Wheat and  Functional Identification of TaRAV in Salt stress” which is considered for Publication in IJMS journal. Article presents new interesting insights in slat stress but article has also several problems. Specific comments to the manuscript I present below:

1.       General comment (les problematic)

Authors must point by point check IJMS publication rules because in many points article is prepared in sloppy way. Any of 51 positions in reference list is not prepared according journal rules. Is little strange that authors did not check this before submission.

Much more problematic:

2.       Introduction section

This part should include or end precisely formulated aim and/or hypothesis of research (as IJMS rules says). Currently manuscript has any of it. It must be improved

3.       Results

One of the most problematic section. In many parts of manuscript Figure captions the errors in spelling or in edition are present. I do not know how it is possible that authors did not check it before submission. Other problems is that that some figures is overloaded with data with makes them difficult to read or make them low quality like Figure 2. Figure 7 must be reformed to show whole scale of expression not parts 0-1.5 and then 100-250. But the most problematic is Figure 4 which apparently should show subcellular localization TaRAV1 protein.  And this is extremely strange and bad approach. Subcellular localizations are localization inside cell on the level of highest magnifications in transmission electron microscope (magnification 18000 and higher) when we see localization of proteins inside or near particular cell organelles or cell compartment which clear ultrastructure could be seen. In this context the used  by authors statements suggest that authors are not aware what they exactly showed or performed. If authors want to presents subcellular localization in truly way the must performed immunogold labeling of TaRAV1 protein and performed analyzes on cross sections in TEM. Currently is illogical also to show localization of TaRAV1 protein in confocal in epidermis of leaf (by the way authors also did not see where exactly localization is present in figure descriptions) in salt stress. If localization with use of GFP is needed to show tissue localization in context the salt stress then it should be performed on cross section where all elements of anatomy of leaf could be seen, where authors will identified tissues in which TaRAV1 protein is truly presents. In slat stress very important is also structure of tissues especially vascular tissues.

4.       Materials and methods:

All used addresses of websites should be cited as reference with number not as http addresses in “()”

Reviewer 2 Report

The work entitled "Genome-Wide Analysis of the RAV Gene Family in Wheat and Functional Identification of TaRAV1 in Salt stress" is well written and presents an interesting research issue. I just need a few changes that I listed below:

Line 30-31 explain the abbreviations as AP2 / ERF, bZIP, MYB, 30 WRKY

Is it necessary to write in block letters for explanations of abbreviations on lines 41-44, since this information is provided in the normal way throughout the text?

Figure 2.- please move the signature of the vertical axis away from the scale, in the present state the captions overlap. Additionally, is it possible to standardize the scale on all presented charts? This will allow a better comparison of the presented data between individual charts. Please also explain what # means in the graphs, as this information was missing in the figure caption.

Figure 3- Please explain the statistical significance in the caption under the picture and explain the # in the graphs.

Figure 5- Please unify the descriptions of the panels a and b in the figure caption on (a) and (b). Please also explain what + means and what - in figure b.

Figure 6- - Please unify the descriptions of panels a and b in the figure caption on (a) and (b). please explain the statistical significance in the caption under the picture and explain # in the graphs. Please provide the scale on the panel b.

Figure 7- Please explain the statistical significance in the caption under the picture and explain the # in the graphs. Additionally, is it possible to standardize the scale on all presented charts? Separately for the first SOS panel, and separate for the RDA panel.

Please delete section 6- 6. Patents

Supplemantary materials - please give the names of individual figures in files (S1-S4), it will allow for better tracking and understanding. Additionally, please provide the scale for S4.